# Dual-task VAE for Node-Level Data Augmentation

## Abstract

Graph Neural Networks (GNNs) have shown great promise in processing graph-structured data, but they often require large amounts of labeled data and are sensitive to noise. In this paper, we propose a novel node-level data augmentation approach that leverages a Variational Autoencoder (VAE) within a dual-task learning framework to address these challenges. Our method utilizes the VAE to generate enriched node representations that capture both structural and feature-related information, which are then combined with the original node features for classification by a Graph Attention Network (GAT). Experiments conducted on the Cora, Citeseer, and Pubmed datasets show that our approach outperforms baseline models, achieving up to 7.3% higher accuracy in Pubmed, and surpassing recent state-of-the-art data augmentation techniques. This work highlights the effectiveness of dual-task learning for robust feature enhancement and advances data augmentation strategies in GNNs.

## 1 Introduction

Graph-structured data is increasingly prevalent across domains, including social networks, biological systems, and recommendation engines. Graph Neural Networks (GNNs) have become central tools for analyzing such data due to their success in tasks like node classification, link prediction, and community detection (Kipf & Welling, 2017; Veličković et al., 2018). Despite this success, GNNs often require extensive labeled data and can be sensitive to noise or structural perturbations, limiting their applicability in settings where high-quality labeled data is scarce or noisy.

Traditional augmentation techniques, such as edge manipulation or node feature masking, aim to increase data diversity and robustness but may fail to fully capture the complex dependencies in graph structures. These methods risk introducing unrealistic modifications that disrupt graph integrity, thus necessitating more refined augmentation approaches (You et al., 2020; Rong et al., 2020).

Variational Autoencoders (VAEs) (Kingma & Welling, 2014) offer a probabilistic framework for learning expressive latent representations and have been adapted for graph tasks like link prediction and graph generation (Kipf & Welling, 2016; Salha et al., 2019). However, their potential for node-level data augmentation, particularly in supervised learning, remains underexplored. Leveraging VAE-generated latent representations within a GNN framework may enrich node features in a way that maintains structural coherence and improves robustness to noise.

In this work, we propose a novel node-level data augmentation method that combines a VAE with a dual-task learning framework to generate enriched node representations. Unlike traditional approaches, our method uses a multi-channel encoder that treats various GNN architectures as complementary filters. Each GNN channel—such as GCN, GAT, SAGE, or GIN—extracts unique structural patterns, effectively decomposing data into multi-faceted representations. This modular, filter-based design allows our framework to flexibly incorporate additional GNN variants, enhancing feature diversity and task adaptability.

Our approach simultaneously trains the VAE for both data reconstruction and node classification, creating latent representations that are both structurally informative and task-relevant. This study is constrained by limited resources, which directs our focus towards methods that can demonstrate robustness and scalability within these constraints. In this way, the VAE serves as a core innovation in generating new features that improve robustness against noise and enriches the original feature

set. By using this combination of VAE-driven feature augmentation and a multi-channel encoder, our framework is not only robust to noisy environments but also highly adaptable to different graph structures, enabling users to select channels based on dataset characteristics and task needs.

Our main contributions are summarized as follows:

- **VAE-based Node-Level Augmentation**: We introduce a VAE framework that produces enriched latent node representations, addressing both data scarcity and robustness in noisy environments.

- **Filter-based Multi-Channel Encoder for Structural Diversity**: By treating multiple GNN architectures (GCN, GAT, SAGE, GIN) as filters that capture distinct structural patterns, our encoder flexibly decomposes data to improve representational quality.

- **Dual-task Learning Framework**: The dual-task approach combines data reconstruction and node classification, yielding a discriminative latent space that enhances node classification while preserving structural integrity.

The remainder of the paper is organized as follows: Section 2 reviews related work in graph data augmentation and VAEs for graphs; Section 3 details our proposed method; Section 4 presents experimental results and analysis; Section 5 discusses findings and limitations, including a discussion on potential applications for diverse graph structures; and Section 6 concludes with future research directions.

## 2 RELATED WORK

### 2.1 GRAPH NEURAL NETWORKS

Graph Neural Networks (GNNs) have become the standard approach for learning on graph-structured data (Kipf & Welling, 2017; Veličković et al., 2018). Key architectures, including Graph Convolutional Networks (GCN) (Kipf & Welling, 2017), Graph Attention Networks (GAT) (Veličković et al., 2018), GraphSAGE (Hamilton et al., 2017), and Graph Isomorphism Networks (GIN) (Xu et al., 2019), have shown effectiveness in tasks such as node classification, link prediction, and community detection. Each of these architectures captures different aspects of graph structure: GCNs focus on local aggregation, GATs use attention mechanisms for adaptive neighbor importance, SAGE aggregates neighborhood information to capture long-range dependencies, and GIN improves expressive power for isomorphism properties in graphs.

However, GNNs often suffer from limitations like over-smoothing—where node features become indistinguishable in deeper layers—and the need for substantial labeled data to achieve high performance (Alon & Yahav, 2021; Zhao et al., 2023). Moreover, single-architecture approaches may be insufficient to fully capture diverse structural information in complex graph data. Our multi-channel encoder addresses these limitations by treating each GNN architecture as a distinct filter, combining their unique strengths in a modular framework to enhance feature diversity and adaptability.

### 2.2 VARIATIONAL AUTOENCODERS FOR GRAPHS

Variational Autoencoders (VAEs) (Kingma & Welling, 2014) provide a probabilistic approach to learning latent representations and have been leveraged in graph learning for tasks such as link prediction and graph generation. Notable VAE-based models, such as VGAE (Kipf & Welling, 2016) and GraphVAE (Simonovsky & Komodakis, 2018), primarily focus on unsupervised learning and graph generation by modeling distributions over adjacency matrices and node features. While these models contribute to generative tasks, their potential for direct node-level data augmentation in supervised learning remains underexplored.

The VAE model is typically trained by minimizing a combined objective of reconstruction loss and Kullback-Leibler (KL) divergence:

$$\mathcal{L}\text{VAE} = \mathbb{E}q_\phi(\mathbf{z}|\mathbf{x})[\log p_\theta(\mathbf{x}|\mathbf{z})] - \text{KL}(q_\phi(\mathbf{z}|\mathbf{x})||p_\theta(\mathbf{z})) \tag{1}$$

where x represents the input node features, and z is the latent representation learned by the encoder. In our work, this objective is adapted to create task-relevant, augmented node representations for supervised node classification, enhancing feature richness and robustness against noise.

## 2.3 DATA AUGMENTATION IN GRAPHS

Data augmentation techniques in graph learning aim to improve model generalization by artificially increasing data diversity. In graph settings, common methods like DropEdge (Rong et al., 2020)—which randomly removes edges—and GraphMix (Verma et al., 2021)—which creates mixed node features—have been proposed to address issues like over-smoothing and overfitting. GraphCL (You et al., 2020) further introduces contrastive learning with augmentations like node dropping and edge perturbation to encourage model robustness.

However, these techniques often rely on random perturbations, which may inadvertently disrupt essential structural information. Our approach differs by generating structured, task-relevant representations using VAE to maintain graph integrity, offering a more refined augmentation strategy that preserves important structural dependencies for node classification.

## 2.4 MULTI-TASK LEARNING IN GNNS

Multi-task learning (MTL) (Caruana, 1997) is commonly used to enhance model generalization by simultaneously training on related tasks, as seen in applications like node classification combined with link prediction (Zhang & Chen, 2018) or community detection (Sun et al., 2019). Dual-task learning, a subset of MTL, enables GNNs to learn more robust and discriminative representations by balancing information across tasks. In our framework, we integrate dual-task learning within the VAE, simultaneously training for both data reconstruction and node classification. This approach improves feature representation quality and robustness, as it allows the model to learn a latent space that benefits both reconstruction and task-specific objectives.

## 2.5 OUR CONTRIBUTION

While VAEs, data augmentation, and multi-task learning have each been explored within GNN frameworks, their combined potential in a modular framework for node-level data augmentation is less explored. By introducing a VAE with a dual-task learning framework and a filter-based multi-channel encoder, we bridge this gap, enabling the generation of enriched, task-relevant node representations. This method is not only effective for node classification but also provides a highly adaptable framework for diverse graph tasks by allowing the selection of different GNN channels based on the data characteristics and task requirements.

## 3 METHODS

Our proposed method consists of two main components: a VAE with a multi-channel encoder for node-level data augmentation, and a GAT for node classification using the augmented features to evaluate the effectiveness of the augmentation process. The **dual-task learning framework** trains the VAE simultaneously for both data reconstruction and node classification, ensuring that the learned representations are both robust and task-relevant.

### 3.1 BASELINE MODEL: DOUBLE-LAYER GAT

To establish a benchmark in our experimental study, we implemented a two-layer GAT as the baseline, capturing relational dynamics within the graph structure for progressive refinement of node representations. After systematic hyperparameter tuning, we identified optimal settings: a learning rate of 0.01, a hidden layer size of 1, 28 attention heads in the first layer, and 12 in the second, using the Adam optimizer with a weight decay of 0.001. Our baseline model achieved a node classification accuracy of 82.8%, with precision, recall, and F1 scores of 0.810, 0.841, and 0.822, respectively, providing a strong foundation for comparisons with our proposed framework.

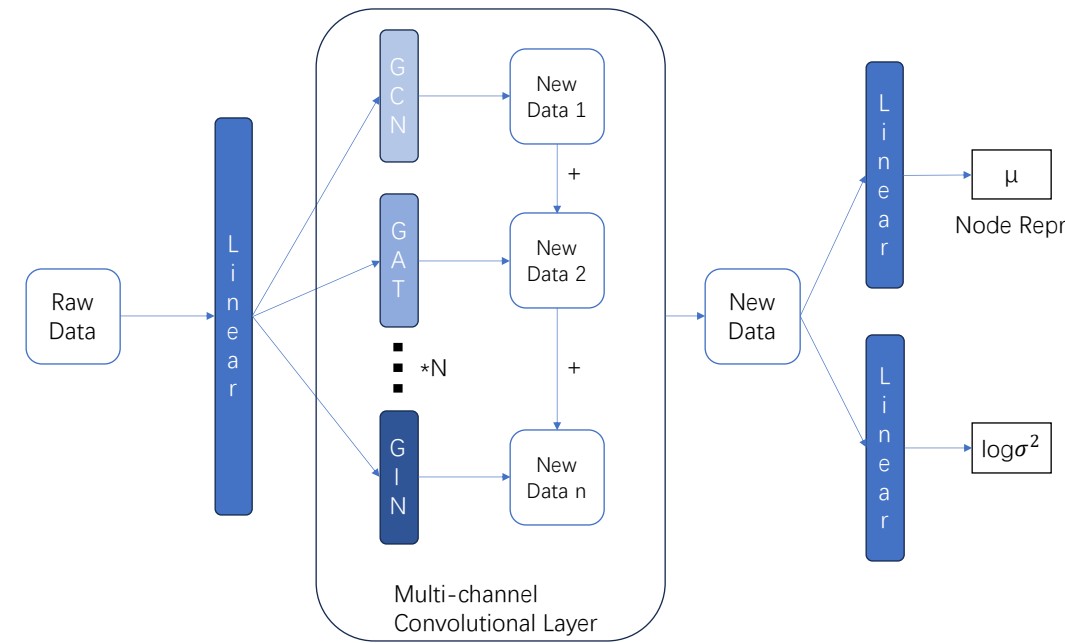

Figure 1: Encoder Structure

## 3.2 UPSTREAM TASK: VAE WITH AUXILIARY NODE CLASSIFICATION TASK

For the upstream task, we designed a **Variational Autoencoder (VAE)** to capture latent representations of the Cora dataset through a dual-task training approach. The VAE encoder maps the input graph data to a latent space, while the decoder reconstructs the data from these latent representations. Notably, the encoder's reparameterized output (specifically the mean $\mu$) serves a dual role: it contributes to the data reconstruction and simultaneously acts as input for an auxiliary node classification task through two stacked linear layers. This dual-task setup allows the VAE to produce embeddings that are robust to noise while enhancing downstream classification performance.

### 3.2.1 ENCODER ARCHITECTURE

The encoder design, shown in Figure 1, captures features and structures through several key stages:

**- Initial Linear Layer**   Node features are first projected into a higher-dimensional space, providing an enriched representation that supports subsequent convolutional operations.

**- Multi-channel Convolutional Layers**   The encoder's core component is the multi-channel convolutional layer, comprising multiple parallel graph convolutional operations. By treating GCN, GAT, SAGE, and GIN layers as unique filters, each capturing different structural properties, the model gains a more comprehensive understanding of the graph:

- **GCN Layer**: Aggregates local connectivity patterns to capture neighborhood features.
- **GAT Layer**: Uses an attention mechanism to prioritize important nodes, enhancing relational representation.
- **SAGE Layer**: Aggregates information for long-range dependencies, offering a broader graph perspective.
- **GIN Layer**: Emphasizes isomorphism properties, maintaining nuanced node feature representations.

**- Feature Fusion**   The outputs of the convolutional layers are concatenated, merging the strengths of each graph convolutional operation into a single, enriched feature representation.

**- Output Linear Layers**   This concatenated representation is then fed into two parallel linear layers to estimate the latent space parameters $\mu$ (mean) and $\log \sigma^2$ (log variance), defining the latent distribution for reconstruction.

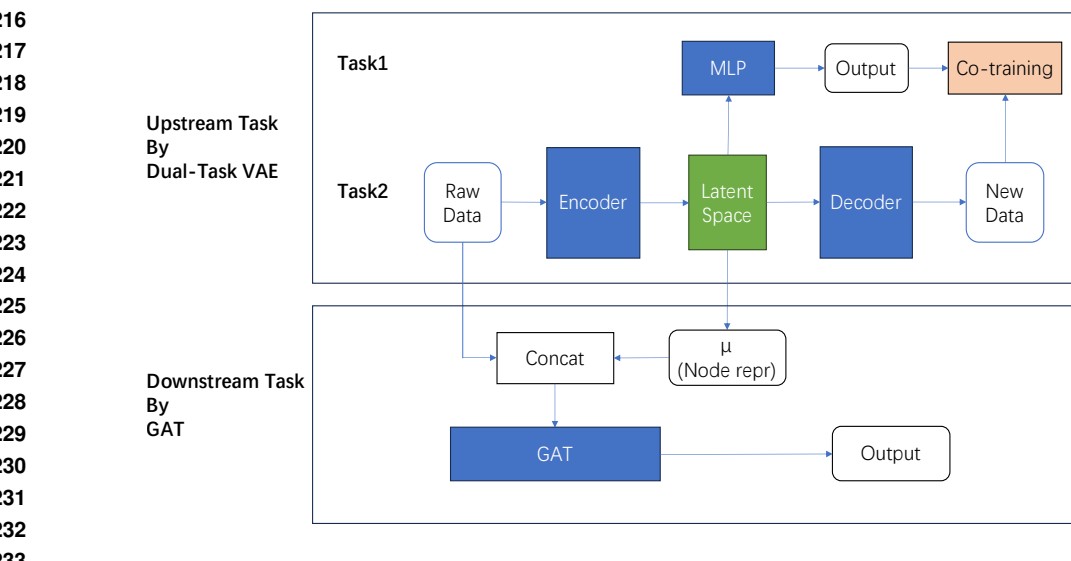

Figure 2: Two-stage Experimental Framework

### 3.2.2 VAE TRAINING

**VAE Loss** The VAE's loss function combines reconstruction loss and a KL divergence term:

$$\mathcal{L}_{\text{VAE}} = \mathbb{E}_{q_\phi(\mathbf{z}|\mathbf{x})}[\log p_\theta(\mathbf{x}|\mathbf{z})] - \text{KL}(q_\phi(\mathbf{z}|\mathbf{x})||p_\theta(\mathbf{z})) \tag{2}$$

where $\mathbf{x}$ represents the input node features, and $\mathbf{z}$ is the latent representation.

**Dual-task Training** In our training framework, the encoder output serves both reconstruction and node classification. By training the encoder on both tasks, it learns to generate latent representations beneficial for both, enhancing classification accuracy and retaining high-quality reconstructions.

**Loss Weight Adjustment** The dual-task loss function incorporates weighted contributions from reconstruction and classification losses:

$$\mathcal{L}_{\text{total}} = a \cdot \mathcal{L}_{\text{recon}} + b \cdot \mathcal{L}_{\text{class}} \tag{3}$$

where $a$ and $b$ are weight parameters for each loss. In practice, the classification loss tends to be **10 to 100 times smaller** than the reconstruction loss. To balance these, we amplify the classification loss by approximately **4500 times**, achieving a scale alignment that prevents dominance by either task, optimizing overall performance and accuracy.

### 3.3 NODE-LEVEL DATA AUGMENTATION STRATEGY

To augment node features, we concatenate the latent vector $\mu$ (obtained via the encoder's reparameterization) with the original node feature vector `data.x`. This combined representation enriches each node's feature vector, enhancing the downstream model's ability to classify nodes accurately.

### 3.4 DOWNSTREAM TASK: GAT WITH AUGMENTED FEATURES

For the downstream task, we use the augmented feature representations as input to a two-layer GAT model. Additional hyperparameter tuning was conducted to optimize performance, validating the effectiveness of our data augmentation strategy.

### 3.5 PERFORMANCE EVALUATION

We evaluate the effectiveness of our proposed method using four metrics: accuracy, F1 score, precision, and recall, calculated on a consistent data split to ensure fair comparison. This setup enables a comprehensive assessment of model performance for node classification tasks.

We compare three configurations: the baseline GAT with original features, GAT with single-task VAE-augmented features, and GAT with features augmented by our dual-task VAE. This comparison highlights the impact of our augmentation strategy and dual-task learning on classification accuracy and model robustness.

## 4 EXPERIMENTAL RESULTS

### 4.1 DATASETS

We evaluated our method on three widely used benchmark citation network datasets: Cora, Citeseer, and Pubmed (McCallum et al., 2000; Giles et al., 1998; Sen et al., 2008). These datasets cover a range of graph sizes, feature dimensions, and sparsity levels, making them ideal for testing the robustness and generalizability of graph-based models.

- **Cora**: 2,708 nodes, 5,429 edges, 1,433 features, 7 classes.
- **Citeseer**: 3,327 nodes, 4,732 edges, 3,703 features, 6 classes.
- **Pubmed**: 19,717 nodes, 44,338 edges, 500 features, 3 classes.

### 4.2 EXPERIMENTAL FRAMEWORK

Our study adopts a two-stage framework to enhance GNN performance for node classification tasks. In the first stage, a Variational Autoencoder (VAE) is used to learn latent representations of the graph data, capturing both structural and feature information. In the second stage, these latent representations are combined with raw features to serve as inputs for a Graph Attention Network (GAT), enabling enriched feature-based classification. The VAE is trained under a dual-task learning framework to ensure that the learned representations are both task-relevant and structurally coherent.

### 4.3 EXPERIMENTAL SETUP

Due to computational resource constraints, this study evaluates the proposed method on three widely used benchmark datasets: Cora, Citeseer, and Pubmed. While these datasets are smaller in scale compared to emerging large-scale graph benchmarks, they provide a well-established foundation for validating methodological effectiveness. Future work will explore the scalability of the proposed framework on larger and more complex datasets as resources permit.

We follow the dataset splits used in (Yang et al., 2016), with 20 nodes per class for training, 500 nodes for validation, and 1,000 nodes for testing. All models were implemented in PyTorch and PyTorch Geometric (Fey & Lenssen, 2019), and hyperparameter tuning was performed on the validation set.

The following experimental settings were adopted: Random seed: 42, Optimizer: Adam, Learning rate: 0.0001, Weight decay: 0 and Loss weight adjustment: Classification loss scaled by a factor of 4,500 to align it with the reconstruction loss magnitude.

To evaluate the proposed method's robustness and effectiveness, we conducted experiments under two conditions:

- **Fixed conditions**: The same random seed (42) was used for both augmented data generation and model training to ensure consistency and highlight the method's potential.
- **Random conditions**: Different random seeds were used for augmented data generation and training across multiple runs, reflecting the method's performance in varying real-world scenarios.

This dual evaluation framework allows for a comprehensive assessment of both the method's peak performance and its robustness across diverse settings.

### 4.4 IMPACT OF VAE NODE-LEVEL DATA AUGMENTATION

Single-source configurations (e.g., Decoder-only: 80.7% accuracy on Cora) showed limited performance. Combined configurations (Raw+NR) significantly improved accuracy (up to 88.6% under MCC), leveraging latent features and preserving raw structural information.

Table 1: Ablation Study on Cora Dataset with MCC: GCN+GAT+SAGE+GIN

| Model | Train Data | Dual Task | Loss Adjust | Accuracy (%) | F1 (%) |
|---|---|---|---|---|---|
| GAT+DVAE | Decoder (VAE-Only) | False | False | 80.7 | 79.9 |
| GAT+DVAE | NR (Latent Only) | False | False | 81.8 | 81.1 |
| GAT+DVAE | Raw+NR | False | False | 82.6 | 81.9 |
| GAT+DVAE | Raw+NR | True | False | 82.6 | 82.2 |
| GAT+DVAE | Raw+NR | True | True | **88.6** | **87.3** |

## 4.5 EFFECT OF DUAL-TASK TRAINING AND LOSS WEIGHT ADJUSTMENT

The dual-task framework demonstrated measurable benefits for node classification, increasing accuracy by approximately 0.6% on the Cora dataset when enabled. Furthermore, scaling the classification loss by a factor of 4,500 to align its magnitude with the reconstruction loss significantly boosted performance, achieving an accuracy of 88.6% on the Cora dataset. This highlights the importance of task-relevant latent representations and balanced optimization in node classification tasks.

## 4.6 EFFECT OF MCC ARCHITECTURE

To evaluate the impact of the multi-channel convolutional layer (MCC) on model performance, we conducted a detailed ablation study. Starting from a single GCN layer, we progressively added more GNN variants (GAT, SAGE, GIN) to construct the MCC architecture. The results, shown in Table 2, demonstrate that incorporating additional GNN variants consistently improves performance. This improvement can be attributed to the diverse structural patterns captured by different GNN layers, with GIN effectively mitigating over-smoothing and SAGE capturing long-range dependencies.

Table 2: MCC Structure Influence on Results

| MODEL | ARCHITECTURE | ACC | F1 |
|---|---|---|---|
| GAT+DVAE | MCC: GCN+GAT | 0.865 | 0.849 |
| GAT+DVAE | MCC: GCN+GAT+SAGE | 0.878 | 0.860 |
| GAT+DVAE | MCC: GCN+GAT+SAGE+GIN | **0.886** | **0.873** |

## 4.7 COMPARISON OF AUGMENTATION METHODS ON GRAPH DATASETS

In Table 3, we present a comparative analysis of the performance of GAT+DVAE against the baseline model, two-layer GAT, across various datasets, demonstrating the general effectiveness of our approach with different random seeds. Building on this overview, Table 4 delves deeper into the specifics of our method's performance on the Cora dataset, where GAT+DVAE is pitted against other state-of-the-art graph augmentation techniques. Key observations include:

- **Baseline and Traditional VAE Usage (Decoder-Only)**: The GAT+Decoder (VAE-Only) configuration, representing a traditional use of VAE for data generation, achieves an accuracy of 80.7%. While this result demonstrates the utility of decoder-generated features, it is lower than methods that integrate latent representations or task-specific features.

- **Supervised and Self-Supervised Augmentation Methods**: Recent methods like DropEdge and GraphMAE leverage self-supervised learning or edge perturbations for augmentation (Hou et al., 2022). GraphMAE achieves 84.2%, while DropEdge reaches 87.6%, showing their ability to address over-smoothing and improve generalization.

- **Our Method (GAT+DVAE)**: By combining task-relevant latent features, raw features, and dual-task training, GAT+DVAE achieves the highest accuracy of 88.6%, outperforming GraphMAE (+4.4%) and DropEdge (+1.0%). This highlights the advantages of our framework in integrating structural and task-relevant information.

Table 3: Performance Comparison Across Fixed and Random Conditions

| Dataset | Condition | Model | Accuracy (%) | Std (%) | Improvement (%) |
|---------|-----------|-------|--------------|---------|-----------------|
| Cora | Random | GAT | 83.0 | ±0.7 | - |
| | Fixed | GAT+DVAE | 88.1 | ±0.4 | +6.1 |
| | Random | GAT+DVAE | 88.1 | ±0.3 | +6.1 |
| | | | | | |
| Citeseer | Fixed | GAT | 70.1 | ±0.8 | - |
| | Fixed | GAT+DVAE | 75.4 | ±0.8 | +7.6 |
| | Random | GAT+DVAE | 74.1 | ±1.5 | +5.7 |
| | | | | | |
| Pubmed | Fixed | GAT | 79.0 | ±0.3 | - |
| | Fixed | GAT+DVAE | 85.7 | ±0.2 | +8.5 |
| | Random | GAT+DVAE | 85.6 | ±0.7 | +8.4 |

**Note1:** Fixed conditions use the same seed (42) to generate augmented data and run the experiments in different seeds, ensuring consistency. Random conditions involve varying seeds for both data generation and training, reflecting the method's robustness across different settings.

**Note2:** The accuracy for the GAT on Citeseer is lower than 72.5% in (Hou et al., 2022), due to the use of the two-layer GAT architecture in our experiments.

Table 4: Comparison of Augmentation Methods on Cora Dataset

| Model | Accuracy (%) | Model | Accuracy (%) |
|-------|--------------|-------|--------------|
| **Unsupervised Methods** | | | |
| GAE | 71.5 ± 0.4 | GPT-GNN | 80.1 ± 1.0 |
| GATE | 83.2 ± 0.6 | DGI | 82.3 ± 0.6 |
| MVGRL | 83.5 ± 0.4 | GRACE | 81.9 ± 0.4 |
| BGRL | 82.7 ± 0.6 | InfoGCL | 83.5 ± 0.3 |
| CCA-SSG | 84.0 ± 0.4 | GraphMAE | 84.2 ± 0.4 |
| | | | |
| **Supervised Methods** | | | |
| GAT | 83.0 ± 0.7 | GAT+partitioning | 80.11 ± 0.84 |
| GAT+Decoder (VAE-Only) | 80.7 ± 0.5 | GCN | 81.5 ± 0.7 |
| GAT+completion | 80.5 ± 1.2 | GCN+DropEdge | 87.6 |
| GAT+clustering | 79.4 ± 0.7 | **GCN+DVAE (Our)** | **87.9 ± 0.4** |
| **GAT+DVAE (Our)** | **88.1 ±0.4** | | |

## 4.8 VISUALIZATION OF LATENT SPACE

To validate the quality of the VAE-learned representations, we visualized the latent embeddings using t-SNE, as shown in Figure 3. The augmented representations exhibited clear class boundaries, illustrating the improved distinguishability of node features after augmentation.

## 5 DISCUSSION

### 5.1 EXPERIMENTAL RESULTS ANALYSIS AND ABLATION STUDY

The results show that our framework greatly improves GAT's performance in node classification. The ablation study revealed that combining raw features with latent representations (Raw+NR) significantly outperformed single-source methods, with accuracy on Cora increasing from 80.7% to 88.6%. This underscores the value of task-relevant latent features and confirms the effectiveness of our dual-task framework in balancing reconstruction and classification goals.

### 5.2 IMPACT OF ARCHITECTURE COMPLEXITY

Our experiments indicate that increasing architectural complexity by integrating various GNNs (GCN, GAT, SAGE, GIN) into the multi-channel convolutional layer (MCC) enhances performance. Each GNN contributes unique strengths, such as GIN's ability to prevent over-smoothing and SAGE's capacity for long-range dependency capture, leading to more robust node embeddings. This suggests that Further expanding MCC could enhance performance.

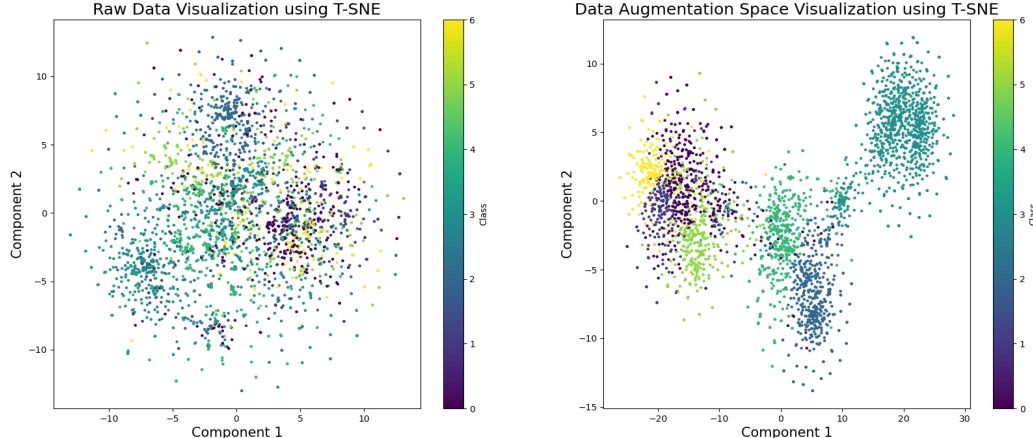

Figure 3: t-SNE Visualization of Latent Space Embeddings

## 5.3 Effectiveness of Dual-Task Training and Loss Weight Adjustment

The dual-task learning framework, combining data reconstruction and node classification, proved essential for creating task-relevant latent spaces. Without dual-task training, the learned latent representations primarily reflect the reconstruction objective, limiting their utility for downstream tasks. Enabling dual-task training increased accuracy on the Cora dataset by approximately 0.6%, while further loss weight adjustment (scaling classification loss by 1,000x) aligned the two objectives, resulting in a final accuracy of 88.6%. This approach effectively balances the competing objectives, enabling the model to generate task-relevant features that maintain structural coherence.

## 5.4 Mitigating Over-smoothing with Node-Level Data Augmentation

Over-smoothing, a well-known challenge in GNNs, occurs when node representations become indistinguishable in deeper networks. Our method's node-level data augmentation strategy effectively addresses this issue by introducing enriched features derived from the VAE's latent space. By concatenating raw features with task-specific latent representations, our approach preserves node-level distinctions, enabling GAT to achieve higher classification accuracy and improved robustness. This enhancement is particularly evident in models incorporating multi-channel convolutional layers, which capture diverse local and global structural patterns.

## 5.5 Visualization and Interpretability

Visualization of the VAE-learned latent space further validates the model's ability to improve node-level feature distinguishability. As shown in Figure 3, the t-SNE visualization reveals well-separated class boundaries, indicating that the augmented features enhance class separability. This interpretability is crucial for understanding how the model processes complex graph-structured data and demonstrates that the learned representations align with the underlying class structure. Such visualization provides valuable insights for analyzing and refining node embeddings.

## 5.6 Future Applications and Task Generalization

The flexibility of our VAE-based data augmentation framework makes it adaptable to a wide range of graph-related tasks. By modifying the auxiliary task in the dual-task framework, the model can generate latent representations tailored to applications such as community detection, link prediction, and graph clustering. Additionally, expanding the MCC architecture to include more specialized GNN variants could further enhance feature expressiveness, enabling the framework to generalize across diverse graph datasets. For instance, integrating hierarchical GNNs or relational GNNs could improve performance on multi-relational or hierarchical graphs.

### 5.7 LIMITATIONS AND FUTURE WORK

#### 5.7.1 RESOURCE CONSTRAINTS AND PRACTICAL FEASIBILITY

This study was conducted under constrained computational resources, which limited the scale of experiments to medium-sized datasets. Despite these constraints, the proposed framework achieved state-of-the-art performance, demonstrating its efficacy in resource-limited environments. Future research will aim to extend the evaluation to large-scale datasets, such as those in the Open Graph Benchmark (OGB), and explore efficient model optimization techniques to enhance scalability.

#### 5.7.2 COMPUTATIONAL COMPLEXITY AND SCALABILITY

A notable limitation of our method is the increased computational complexity introduced by the VAE and multi-channel encoder. Training the VAE with dual-task objectives requires additional computational resources, particularly for large-scale graphs. Future research could explore lightweight convolutional layers, model pruning techniques, or efficient training algorithms to address this challenge. Transfer learning and self-supervised learning could also reduce dependence on labeled data, making the framework more scalable and applicable to real-world scenarios.

#### 5.7.3 GENERALIZABILITY TO LARGER AND NOISIER GRAPH DATASETS

While our method performs well on Cora, Citeseer, and Pubmed datasets, its effectiveness on larger or noisier graphs remains to be validated. Graphs with complex structures, such as dynamic or hierarchical graphs, may require architectural modifications, such as adaptive latent space modeling or dynamic feature fusion mechanisms. Future experiments on diverse datasets, including social networks or knowledge graphs, will further evaluate the framework's robustness and generalizability.

#### 5.7.4 INTEGRATION WITH OTHER DATA AUGMENTATION TECHNIQUES

Although our study focuses on VAE-based node-level augmentation, integrating other augmentation techniques could further enhance model performance. For example, combining edge perturbation, subgraph sampling, and contrastive learning with our method could create a hybrid augmentation framework. This approach would generate more diverse and task-specific data variations, enabling the model to adapt to a broader range of graph analysis tasks.

#### 5.7.5 POTENTIAL FOR BROADER APPLICATIONS

The flexibility of our framework extends beyond node classification. For instance, by adapting the dual-task framework to optimize for link prediction or community detection, the model could address diverse graph analysis challenges. Future work could explore multi-task configurations that combine these objectives, enhancing the framework's utility for multi-faceted graph analytics.

## 6 CONCLUSION

This study presents a novel VAE-based data augmentation method that significantly enhances GNN performance on node classification tasks. By integrating multi-channel convolutional layers and a dual-task training framework, we developed a robust approach for managing noisy and incomplete data, achieving notable improvements in classification accuracy and feature distinguishability.

The adaptability of this framework extends beyond node classification to other graph-based tasks, such as community detection and link prediction, by adjusting the auxiliary task in the dual-task learning setup. Future research could explore incorporating more advanced architectures, optimizing for larger datasets, and integrating additional data augmentation techniques to further enhance the model's effectiveness and scalability.

Overall, this VAE-based augmentation framework offers a promising direction for constructing flexible and high-performance models in graph data analysis, contributing to the development of robust and adaptable solutions for various applications in the graph learning domain.

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
