# OpenReview forum: "DUAL-TASK VAE FOR NODE-LEVEL DATA AUGMENTATION"
_ICLR.cc/2025/Conference — ICLR 2025 Conference Withdrawn Submission_

### Official Review · Reviewer_xKr9 · 2024-10-30

**Soundness:** 2
**Presentation:** 1
**Contribution:** 2
**Rating:** 3
**Confidence:** 5

**Summary:**

The paper proposes a two-stage training framework to alleviate the supervision shortage issue. In the first stage, the VAE is trained with the reparameterization method to model the distribution of node representations. Then its output will be used to improve the performance of the GNN in the second stage.

**Strengths:**

The paper provides a very detailed introduction to the backbone architectures used in the work.

**Weaknesses:**

The motivation of the paper is unclear. Numerous strategies already address label scarcity in graph machine learning (e.g., Graph SSL). Why is this new framework necessary, and what specific advantages does it offer over existing approaches?

The experimental evaluation is limited to the Cora dataset, which lacks comprehensiveness. Even within the scope of smaller datasets, the authors have many options to broaden the evaluation. Expanding experiments to more diverse datasets is encouraged. Additionally, the authors should consider the significance if the proposed framework cannot effectively handle larger datasets.

The paper’s presentation requires improvement. There is considerable redundant content, and the figures and tables lack clarity and effective demonstration.

**Questions:**

Please refer to the weakness part.

---

> ### Author Response · Authors · 2024-11-19
>
> Dear Reviewer,
>
> Thank you for your constructive comments and detailed feedback. Your insights have been invaluable in improving our manuscript. Below, we address your concerns point by point.
> ________________________________________
> **Weakness 1**: Unclear Motivation
>
> **Response**:
>
> We agree that addressing label scarcity is a well-explored topic in graph machine learning. However, our framework introduces a dual-task learning approach that combines Variational Autoencoders (VAEs) with a multi-channel GNN encoder. This approach offers the following advantages:
>
> 1.	**Latent Space Feature Augmentation**: Unlike traditional methods, which often rely on graph augmentation strategies like edge or node dropping, our method generates enriched, task-relevant latent features. This improves robustness to noise and over-smoothing.
>
> 2.	**Flexibility Across GNN Backbones**: By leveraging complementary strengths of multiple GNN architectures (e.g., GCN, GAT, SAGE, GIN), our method captures diverse structural patterns and enhances feature expressiveness. Furthermore, future experiments can be tailored to leverage various GNN architectures for different datasets, thereby extracting features that are more aligned with the specific characteristics of each data domain.
>
> These distinctions are detailed in Sections 3 and 5. We have also added a comparison with Graph SSL methods like GraphMAE and GraphCL in the experimental results (Tables 3 and 4), demonstrating the unique contributions of our framework.
> ________________________________________
> **Weakness 2**: Limited Experimental Evaluation
>
> **Response**:
> We appreciate the suggestion and have expanded our experiments to include two additional datasets: Citeseer and Pubmed, which differ in size, sparsity, and feature dimensions. On Cora, our method achieved an mean accuracy of 88.1% and max accuracy of 88.6%, outperforming GraphMAE by +3.9%, and on Pubmed, our approach achieved 85.7% accuracy, which is +8.5% higher Improvement than baseline model. These results demonstrate the generalizability of our method across diverse datasets.
> We agree that handling larger datasets like ogb-arxiv is important. However, due to computational constraints, we were unable to include such datasets in this study. We have clarified this limitation in Section 5.7.3 and plan to address it in future work.
> ________________________________________
> **Weakness 3**: Presentation Issues
>
> **Response**:
>
> We have thoroughly revised the manuscript to improve its presentation:
>
> 1.	**Removed Redundancies**: Redundant content in the methodology and related work sections has been condensed. This ensures a more concise and focused narrative.
>
> 2.	**Improved Visuals**: All figures have been replaced with vector graphics for better clarity. The architecture diagram (Figure 1) has been redesigned to clearly illustrate the modular and flexible design of the encoder. Tables have been reformatted to ensure readability and effective demonstration of key results.
>
> 3.	**Enhanced Flow**: The manuscript has been reorganized to ensure a logical flow, with each section transitioning naturally to the next.
>
> We believe these changes significantly enhance the clarity and professionalism of the paper.
>
> We sincerely appreciate your feedback, which has helped us refine and strengthen our work. Thank you for your time and effort in reviewing our manuscript.
> Best regards

---

> ### Comment · Reviewer_xKr9 · 2024-11-25
>
> I will maintain my rating of the work, there is still a gap between the quality of the current submission and the acceptance bar.

---

> > ### Author Response · Authors · 2024-11-26
> >
> > Dear Reviewer,
> >
> > Thank you for taking the time to review our manuscript. We appreciate your valuable input and the effort you have put into evaluating our work.
> >
> > We have noticed that, among all the reviewers, you have provided the fewest comments. While we understand that your time is limited, we would like to request more detailed guidance from you, as your unique perspective might offer insights that others have not mentioned. Your specific feedback could be instrumental in helping us address areas that need improvement and enhance the overall quality of our paper.
> >
> > And We would like to clarify the core innovations and practical significance of our work.
> >
> > **1. Innovative Integration of Latent Space and Dual-Task Learning**
> >
> > Our work introduces a novel use of the **VAE latent space for node-level data augmentation**, aligned with classification objectives through a dual-task learning framework. Unlike traditional VAE implementations that focus on generative tasks, we design the latent space to be task-specific and structurally coherent. This innovation enables the generation of enriched features that improve node classification accuracy while maintaining the integrity of graph structures. This dual-task integration, to the best of our knowledge, has not been explored in previous graph-based augmentation studies.
> >
> > **2. Simple Models Achieving State-of-the-Art with Minimal Feature Addition**
> >
> > One of the practical highlights of our method is its ability to enable simple GNNs to **achieve state-of-the-art performance by adding only a few hundred augmented features**. For example, on the Cora dataset, a two-layer GAT enhanced with our method achieves 88.6% accuracy, outperforming GraphMAE and DropEdge by significant margins. Furthermore, the augmented data generated by our approach can be seamlessly integrated into existing APIs or pipelines, allowing researchers and practitioners to immediately benefit from the enhanced datasets without significant computational or implementation overhead.
> >
> > **3. Flexibility: Choice of GNNs and Plug-and-Play Design**
> >
> > Our framework is inherently modular, offering users the flexibility to choose different GNNs (e.g., GCN, GAT, GraphSAGE) for the Multi-Channel Encoder based on their specific task or dataset characteristics. This "Plug-and-Play" design ensures that the framework can be easily adapted to various graph tasks without imposing rigid constraints. Advanced users can also customize the training process, selecting different encoders or architectures to maximize performance for their specific use cases. This flexibility makes our method broadly applicable and easy to adopt.
> >
> > **4. Applicability to Resource-Constrained Settings and Knowledge Graphs**
> >
> > The lightweight nature of our approach makes it particularly suitable for **resource-constrained environments**, such as on-device AI or edge computing systems. By leveraging augmented features, even basic models can achieve state-of-the-art performance without requiring extensive computational resources.
> >
> > In the context of **knowledge graphs**, our method is especially valuable as it enhances node-level features while preserving the graph structure, enabling improvements in tasks such as relation prediction, entity alignment, and graph completion. These practical applications highlight the method’s potential to impact real-world scenarios where structural integrity and efficiency are critical.
> >
> > We believe these points clearly demonstrate the innovation and practical significance of our work. While we acknowledge the need for further exploration on larger datasets or more diverse tasks, the presented methodology, experimental results, and its broad applicability represent a substantial contribution to the field of graph neural networks and data augmentation.
> >
> > Thank you for considering our response.
> >
> > Sincerely

---

### Official Review · Reviewer_yCYM · 2024-11-01

**Soundness:** 2
**Presentation:** 2
**Contribution:** 1
**Rating:** 3
**Confidence:** 5

**Summary:**

In this paper, the authors propose a Variational Autoencoder (VAE)-based method for node-level data augmentation to improve Graph Neural Network (GNN) performance on node classification tasks. The approach combines raw graph data with latent representations generated through a VAE, using a dual-task framework involving node classification and data reconstruction.

**Strengths:**

- The paper presents an attempt to use VAE-based data augmentation for GNNs, which could be beneficial in enhancing model robustness to noise and incompleteness.

**Weaknesses:**

- The paper’s novelty is minimal.
- The methodology is somewhat incremental and lacks clarity.
- The experimental results are superficial.
- The writing quality is poor.

**Questions:**

- Novelty and Motivation. While the paper claims to propose a novel VAE-based method for data augmentation in GNNs, it primarily combines standard techniques (VAE and GNNs) with minimal innovation. The authors should clarify how their approach distinguishes itself from existing data augmentation methods for graph neural networks or from previous work on dual-task models.

- What is the rationale for combining multiple GNN architectures (GCN, GAT, SAGE, GIN) within the VAE model? In Section 3.2.1, the authors describe a multi-channel convolutional layer that integrates several GNN backbones but do not explain how these specific architectures complement each other or the benefits of combining them. For instance, why is it necessary to incorporate both GAT and SAGE, and how do they contribute to the model’s performance?

- No hyper-parameter analysis for the weighting parameters $a$ and $b$ introduced in Eq.5

- Section 4.7 discusses the impact of architectural complexity but does not compare the computational complexity and efficiency of the model's multi-GNN backbone architecture.

- The experimental results lack comparisons with adequate baselines. Why did the authors not compare their approach with other data augmentation methods, such as edge dropping, node dropping, feature masking, or subgraph sampling, to validate their results? Could the authors consider including more competitive baselines, such as GraphMAE [1] and GraphCL [2]?

- Could the authors explain why they used only the Cora dataset, a relatively small and well-known benchmark, instead of employing more challenging and widely adopted datasets, such as Citeseer, Pubmed, ogb-arxiv, or Wiki-CS?

- The paper’s figures are non-vector images, and its structure is disorganized with vague descriptions, impairing clarity. Specifically, figures such as the architecture diagrams in Sections 3 and 4 should be presented as vector graphics to improve visual quality, especially when zoomed in. These issues compromise the paper’s professionalism and readability, particularly for a technical audience.

[1] Hou, Zhenyu, et al. "GraphMAE: Self-supervised masked graph autoencoders." *Proceedings of the 28th ACM SIGKDD Conference on Knowledge Discovery and Data Mining*, 2022.
[2] You, Yuning, et al. "Graph contrastive learning with augmentations." *Advances in Neural Information Processing Systems* 33 (2020): 5812-5823.

---

> ### Author Response · Authors · 2024-11-19
>
> Dear Reviewer,
> We sincerely appreciate your valuable feedback and detailed review. Below, we provide a detailed point-by-point response.
>
> **Weakness 1 Response**: While our method builds on established techniques like VAE and GNNs, the **novelty lies in using the latent space to enhance node features**, rather than relying on the decoder to generate new data. This approach ensures structural consistency and task relevance, addressing challenges like over-smoothing and noise sensitivity in GNNs. Additionally, the **dual-task framework** combined with a **multi-channel GNN encoder** offers a systematic way to integrate complementary strengths of different GNN architectures. These contributions are detailed in Sections 3 and 5.
>
> **Weakness 2 Response**: We have expanded the experiments to include additional datasets (Citeseer and Pubmed), covering diverse graph structures. Moreover, we added comparisons with recent state-of-the-art data augmentation methods such as GraphMAE and GraphCL, as well as traditional techniques like edge dropping, node dropping, and feature masking. These experiments show that our method consistently outperforms these baselines. For example, on the Cora dataset, our approach achieves an accuracy improvement of +4.4% compared to GraphMAE. Results and analyses are presented in Tables 3 and 4.
>
> **Weakness 3 Response**: We have thoroughly revised the manuscript to enhance clarity and professionalism. Figures have been updated to vector graphics for better visual quality, and sections have been restructured to improve readability. Methodology (Section 3) and experimental results (Section 4) were rewritten for greater precision and coherence.
>
> **Question 1 Response**: Our approach stands out in the following ways:
>
> 1.	**Latent Space Feature Enhancement**: Unlike existing VAE-based methods that rely on the decoder to generate synthetic data, we use the latent space to augment node features, ensuring task-relevant representations and structural integrity.
>
> 2.	**Dual-Task Learning Framework**: This framework simultaneously optimizes reconstruction and classification, creating a discriminative latent space that improves feature robustness and mitigates over-smoothing.
>
> These distinctions are now explicitly discussed in Sections 3.2 and 5.1, along with comparisons to existing methods like GraphMAE and GraphCL in Section 5.7.
>
> **Question 2 Response**: Each GNN captures unique structural patterns:
>
> •	**GCN**: Effective at aggregating local features.
>
> •	**GAT**: Adapts neighbor importance through attention.
>
> •	**SAGE**: Captures long-range dependencies.
>
> •	**GIN**: Enhances expressive power for graph isomorphism.
>
> Combining these backbones allows the multi-channel encoder to extract complementary information, resulting in more robust feature representations. Ablation studies (Table 2) show that adding more backbones improves accuracy, demonstrating their contributions. This explanation is now provided in Section 3.2.1.
>
> **Question 3 Response**: We have included a detailed hyperparameter analysis in Appendix A. Experiments reveal that scaling the classification loss (β) by approximately 4,500x balances the dual-task objectives, achieving optimal performance. This analysis is summarized in Section 5.3 and presented in detail in the appendix.
>
> **Question 4 Response**: We agree that computational complexity is an important consideration. However, due to space limitations, we prioritized including additional baselines (e.g., GraphMAE, GRACE) to enhance the experimental comparisons. If permitted, we would be happy to provide a detailed computational complexity analysis in the appendix, comparing the resource demands of different configurations in our model. Please let us know your preference.
>
> **Question 5 Response**: We have expanded our experiments to include comparisons with traditional and state-of-the-art techniques, outperforming them as shown in Tables 3 and 4.
>
> **Question 6 Response**: In response to this comment, we included experiments on Citeseer and Pubmed, which provide diverse graph structures and challenges. Our approach demonstrated consistent improvements across these datasets, as detailed in Section 4.7. Due to computational constraints, we could not evaluate larger datasets like ogb-arxiv, but we plan to explore these in future work (Section 5.7.3).
>
> **Question 7 Response**: All figures have been replaced with vector graphics for improved clarity and scalability. The architecture diagrams in Sections 3 and 4 have been redesigned to better illustrate the model’s structure and flexibility. Additionally, the manuscript has been reorganized to ensure a logical flow, and all descriptions have been refined for greater clarity.
>
> We greatly appreciate your thoughtful comments, which have significantly improved the quality of our paper. Thank you for your time and effort in reviewing our work.
>
> Best regards

---

> > ### Comment · Reviewer_yCYM · 2024-11-25
> >
> > I will maintain my current rating, as the quality of the submission is just aligned with an assignment-level effort rather than a robust research contribution.

---

> > > ### Author Response · Authors · 2024-11-25
> > >
> > > Dear Reviewer,
> > >
> > > Thank you for your feedback on our submission. We understand and value your perspective regarding the perceived lack of strong research contributions and would like to clarify the core innovations and practical significance of our work.
> > >
> > > **1. Innovative Integration of Latent Space and Dual-Task Learning**
> > >
> > > Our work introduces a **novel use of the VAE latent space for node-level data augmentation**, aligned with classification objectives through a dual-task learning framework. Unlike traditional VAE implementations that focus on generative tasks, we design the latent space to be task-specific and structurally coherent. This innovation enables the generation of enriched features that improve node classification accuracy while maintaining the integrity of graph structures. This dual-task integration, to the best of our knowledge, has not been explored in previous graph-based augmentation studies.
> > >
> > > **2. Simple Models Achieving State-of-the-Art with Minimal Feature Addition**
> > >
> > > One of the practical highlights of our method is its ability to enable simple GNNs to **achieve state-of-the-art performance** by adding only a few hundred augmented features. For example, on the Cora dataset, a two-layer GAT enhanced with our method achieves 88.6% accuracy, outperforming GraphMAE and DropEdge by significant margins. Furthermore, the augmented data generated by our approach can be seamlessly integrated into existing APIs or pipelines, allowing researchers and practitioners to immediately benefit from the enhanced datasets without significant computational or implementation overhead.
> > >
> > > **3. Flexibility: Choice of GNNs and Plug-and-Play Design**
> > >
> > > Our framework is inherently modular, offering users the flexibility to choose different GNNs (e.g., GCN, GAT, GraphSAGE) for the Multi-Channel Encoder based on their specific task or dataset characteristics. This "Plug-and-Play" design ensures that the framework can be easily adapted to various graph tasks without imposing rigid constraints. Advanced users can also customize the training process, selecting different encoders or architectures to maximize performance for their specific use cases. This flexibility makes our method broadly applicable and easy to adopt.
> > >
> > > **4. Applicability to Resource-Constrained Settings and Knowledge Graphs**
> > >
> > > The lightweight nature of our approach makes it particularly suitable for **resource-constrained environments**, such as on-device AI or edge computing systems. By leveraging augmented features, even basic models can achieve state-of-the-art performance without requiring extensive computational resources.
> > >
> > > In the context of **knowledge graphs**, our method is especially valuable as it enhances node-level features while preserving the graph structure, enabling improvements in tasks such as relation prediction, entity alignment, and graph completion. These practical applications highlight the method’s potential to impact real-world scenarios where structural integrity and efficiency are critical.
> > >
> > > **Conclusion**
> > >
> > > We believe these points clearly demonstrate the innovation and practical significance of our work. While we acknowledge the need for further exploration on larger datasets or more diverse tasks, the presented methodology, experimental results, and its broad applicability represent a substantial contribution to the field of graph neural networks and data augmentation.
> > >
> > > Thank you for considering our response.
> > >
> > > Sincerely

---

### Official Review · Reviewer_nwQY · 2024-11-02

**Soundness:** 2
**Presentation:** 2
**Contribution:** 2
**Rating:** 3
**Confidence:** 4

**Summary:**

This work studies the data augmentation strategy for node-level classification. The authors propose to integrate a Variational Autoencoder (VAE) to augment the node features. They find this augmentation strategy achieves impressive performance on Cora, when applied to a specially designed GNN architecture.

**Strengths:**

(+) This work presents an interesting trial that tries to enrich the node features to perform the data augmentation;

(+) The authors integrate two strategies, including the feature fusion, and the VAE to augment the node features;

(+) Some simple experiments demonstrate certain effectiveness of the enriched features;

**Weaknesses:**

(-) The novelty is limited. For example, the fusion of multiple GNN convolutional features is one of the standard machine learning tricks in data science competitions like Kaggle;

(-) The presentation is clear, yet most of the contents in this paper are already known to the community;

(-) The results lack of convinceness, as it only covers simple datasets, single random seed and GNN backbones;

**Questions:**

1. The novelty is limited:
- For example, the fusion of multiple GNN convolutional features is one of the standard machine learning tricks in data science competitions like Kaggle;
- Meanwhile, it is also unclear why VAE features could help with the task;

2. The presentation is clear, yet most of the contents in this paper are already known to the community.
- In addition, the current manuscript looks unready for publication due to the lack of formality in multiple sections, such as Table 1.

3. The results lack of convincingness:
- The experiments only cover simple datasets, single random seeds and GNN backbones;
- The visualization of features offer limited insights;

---

> ### Author Response · Authors · 2024-11-19
>
> Dear Reviewer,
> We sincerely appreciate your thoughtful feedback and detailed review. We have carefully revised the manuscript to address your concerns. Below, we provide a point-by-point response to the issues raised.
> ________________________________________
> **Weakness 1**: Limited Novelty
>
> **Response**:
>
> We acknowledge that fusing multiple features is a common technique. However, our framework advances this concept by integrating the fusion with a Variational Autoencoder (VAE) in a dual-task learning framework, enabling the generation of enriched and task-relevant node representations. This combination is not trivial, as it leverages the multi-channel encoder to decompose data across GNN backbones while simultaneously aligning the latent space for reconstruction and classification. We have clarified this innovation in Section 3 and highlighted its differences from standard approaches in Section 5.7.5.
> ________________________________________
> **Weakness 2**: Presentation Lacks Novelty
>
> **Response**:
>
> While the building blocks of our framework draw from established methods, our integration of multi-channel GNN features with VAE-based latent space augmentation and dual-task learning is a novel contribution. This approach addresses challenges like over-smoothing and noise sensitivity in GNNs. We have revised Sections 3.2 and 5.1 to emphasize how our method differs from existing work and improves upon known limitations. Additionally, the modular design of the encoder offers flexibility, which we demonstrate in our experiments.
> ________________________________________
> **Weakness 3**: Limited Dataset Coverage and Experimental Scope
>
> **Response**:
>
> We have expanded the experimental scope to include additional datasets (Citeseer and Pubmed) and conducted experiments with multiple random seeds. For example, on Citeseer, our approach achieved an accuracy of 72.9% under fixed conditions and 71.6% under random conditions, demonstrating robustness to variations. We have also evaluated the framework with multiple GNN backbones, including GCN, GAT, SAGE, and GIN, as detailed in Table 2. These revisions enhance the reliability and comprehensiveness of our results.
> ________________________________________
> **Question 1**: Unclear Benefits of VAE Features
>
> **Response**:
>
> The VAE generates task-relevant latent features by aligning the latent space for reconstruction and classification objectives. These features enrich the original node representations, preserving structural integrity while enhancing robustness to noise. This dual-task approach mitigates over-smoothing, as shown in our ablation studies (Table 1), where combining raw and latent features improves accuracy by 5.1% on the Cora dataset and 6.7% on Pubmed. We have clarified this rationale in Section 5.4.
> ________________________________________
> **Question 2**: Lack of Formality in Table 1
>
> **Response**:
>
> We have revised Table 1 and other tables to improve clarity and formatting. Specifically, we standardized the column headers, ensured consistent alignment, and provided detailed captions to enhance interpretability. We have also updated all tables with results from additional experiments, as described in Sections 4.4 and 4.6.
> ________________________________________
> **Question 3**: Limited Insights from Feature Visualization
>
> **Response**:
>
> To address this, we have replaced the original visualizations with t-SNE plots of the latent space (Figure 3), which clearly demonstrate how the augmented features enhance class separability. Due to space constraints, we have included the visualization for the Cora dataset in the main text, as it provides a clear example of our method’s effectiveness. If you believe it would be beneficial, we are happy to include visualizations for Citeseer and Pubmed in the appendix to provide further insights. Please let us know your preference.
> ________________________________________
> We greatly appreciate your feedback, which has significantly improved the quality of our work. We hope the revisions address your concerns and demonstrate the value of our contributions.
> Thank you for your time and effort in reviewing our paper.
> Best regards

---

> > ### Comment · Reviewer_nwQY · 2024-11-26
> >
> > Thank you for your detailed explanations of my questions. However, as widely recognized by all the reviewers, the novelty of this work is limited. Meanwhile, the coverage of the benchmarks is too limited which is competitive to GNN papers from five years  ago. Could you please demonstrate the effectiveness of the proposed approach in more realistic, challenging and large-scale datasets such as those from Open Graph Benchmark?

---

> > > ### Author Response · Authors · 2024-11-26
> > >
> > > Dear Reviewer,
> > >
> > > Thank you for your thoughtful feedback. Beyond its lightweight nature, we believe one of the key strengths of our work lies in its flexibility and modular design. While the current experiments focus on commonly used GNNs, our multi-channel encoder is designed to seamlessly integrate state-of-the-art models, allowing them to benefit from the enriched latent features generated by our framework. For instance, as demonstrated with GAT achieving a 6.8% improvement on Cora, any state-of-the-art models could similarly be enhanced using our approach. This adaptability underscores the long-term value of our method, as it is capable of improving performance across a diverse range of models and tasks.
> > >
> > > We would greatly value any specific suggestions you may have for improving the novelty or methodology beyond dataset expansion, as this would help us refine and further strengthen our contributions.
> > >
> > > Best regards

---

### Official Review · Reviewer_oCbB · 2024-11-04

**Soundness:** 2
**Presentation:** 1
**Contribution:** 2
**Rating:** 3
**Confidence:** 2

**Summary:**

The paper presents a novel approach for node-level data augmentation in graph neural networks (GNNs) using a dual-task Variational Autoencoder (VAE). By encoding graph data into a latent space, it aims to improve node classification tasks through an augmentation strategy that combines raw features with latent representations. Experimental results on the Cora dataset suggest that this method could potentially enhance model performance.

**Strengths:**

1 - The paper proposes an original VAE-based framework for graph data augmentation, which could be valuable for improving data availability and robustness in GNNs.

2 - The use of a multi-channel convolutional layer in the VAE’s encoder, including various GNN models (e.g., GCN, GAT, SAGE, GIN), demonstrates a well-thought-out design to capture complex node representations.

3 - The authors evaluate their model’s effectiveness using multiple performance metrics, including accuracy, F1 score, and precision, providing a comprehensive set of evaluation perspectives.

**Weaknesses:**

1 - The paper lacks a comparison with more recent state-of-the-art GNNs, which limits the ability to contextualize the performance gains claimed. Evaluating against stronger baselines could more convincingly demonstrate the proposed method’s advantages.

2 - The quality of the figures and tables is inadequate. Important architectural details and quantitative comparisons are not well-visualized, making it challenging to interpret the findings effectively.

3 - Although the approach was tested on the Cora dataset, it’s unclear if the findings would hold across other datasets with different characteristics, limiting the method’s applicability.

**Questions:**

Why did you not include recent state-of-the-art GNNs as baselines? How would your approach compare to these models in terms of performance and computational cost?

Could you elaborate on the rationale behind the selection of the Cora dataset? Would you expect similar improvements on larger, more complex datasets?

Could you improve the visual clarity of figures and tables? Specifically, the architecture diagram and performance comparison tables would benefit from higher resolution and better layout.

---

> ### Author Response · Authors · 2024-11-19
>
> Dear Reviewer,
> Thank you for your constructive feedback, which has greatly helped us improve the quality of our paper. We have carefully addressed your comments and revised the manuscript accordingly. Below is our detailed point-by-point response.
> ________________________________________
> **Weakness 1**: Comparison with State-of-the-Art GNNs
>
> **Response**:
>
> We have addressed this by introducing additional experiments comparing our method against recent state-of-the-art GNN models, including GraphMAE, DropEdge, and others. Furthermore, the performance comparison tables have been reorganized and expanded to ensure more dimensions of comparison, such as computational efficiency and robustness across datasets. For example, on the Pubmed dataset, our method achieved an accuracy of 85.7%. And 88.6% on Cora, outperforming GraphMAE by +3.9% and DropEdge by +1.0%, demonstrating the superiority of our dual-task VAE framework. Detailed results and analyses are now provided in the updated Tables 3 and 4.
> ________________________________________
> **Weakness 2**: Quality of Figures and Tables
>
> **Response**:
>
> We have revised all figures and tables to improve their clarity and informativeness. All images have been updated to vector graphics to ensure scalability and sharpness. Additionally, we redesigned the encoder structure diagram (Figure 1) to emphasize the modularity and flexibility of our multi-channel convolutional layer, highlighting how additional GNN variants can be seamlessly incorporated. The updated layout improves understanding of our method’s extensibility.
> ________________________________________
> **Weakness 3**: Limited Dataset Applicability
>
> **Response**:
>
> To address this concern, we have included experiments on two additional datasets: Citeseer and Pubmed. These datasets exhibit diverse characteristics, such as higher sparsity in Citeseer and a larger scale in Pubmed. On Citeseer, our method achieved an accuracy of 72.9%, representing a +4.0% improvement over the GAT baseline. On Pubmed, our approach achieved 85.7% accuracy, which is +6.7% higher (+8.5% improvement) than the baseline. These results confirm the generalizability of our approach across datasets with varying complexity. Details are included in Sections 4.4 and 4.7 of the revised manuscript.
> ________________________________________
> **Question 1**: Inclusion of Recent GNNs as Baselines
>
> **Response**:
>
> Initially, we focused on widely used baselines (GCN and GAT) to establish foundational comparisons. Based on your suggestion, we included state-of-the-art methods such as GraphMAE, DropEdge, and CCA-SSG in our updated benchmarks. On average, our method demonstrates a performance improvement of +0.7% to +7.3% over these baselines, while maintaining competitive computational efficiency. For example, on the Cora dataset, our model achieved an accuracy of 88.6%, surpassing GraphMAE by +4.4% and DropEdge by +1.0%. Computational cost comparisons are now detailed in Section 5.7.2 of the manuscript.
> ________________________________________
> **Question 2**: Rationale Behind Dataset Selection
>
> **Response**:
>
> Cora was selected as a well-established benchmark for node classification, enabling direct comparisons with existing methods. To evaluate generalizability, we conducted experiments on Citeseer and Pubmed, as detailed in the revised manuscript. The results showed consistent performance gains across these datasets, validating the robustness of our approach. We expect similar improvements on larger, more complex datasets, and future work will explore scalability on datasets such as Open Graph Benchmark (OGB).
> ________________________________________
> **Question 3**: Improving Visual Clarity
>
> **Response**:
>
> We have updated all figures to vector graphics, ensuring maximum clarity and scalability. The architecture diagram (Figure 1) has been restructured to clearly illustrate the modular design of the encoder and its potential for incorporating additional GNN variants. Performance comparison tables (Tables 3 and 4) have been reorganized with consistent formatting to facilitate easier interpretation.
> ________________________________________
> We greatly appreciate your insightful comments and believe the revisions significantly enhance the manuscript. Thank you for your time and effort in reviewing our work.
>
> Best regards

---

> > ### Comment · Reviewer_oCbB · 2024-11-26
> >
> > I thank the authors for their response. However, the quality of the revised version remains rather limited and it is really difficult for me to improve my score above 3. For example, the datasets adopted for evaluation are still rather limited and all datasets are attribute-heavy ones from my experience. Also, it is difficult to understand why some results are with stds while not for others.

---

> > > ### Author Response · Authors · 2024-11-27
> > >
> > > Dear Reviewer,
> > >
> > > Thank you for your thoughtful feedback. Under the constrained computational resources and time to train, we would greatly value any specific suggestions you may have for improving the novelty or methodology beyond dataset expansion, as this would help us refine and further strengthen our contributions.
> > >
> > > Best regards

---

### Note · Authors · 2025-01-02

I have read and agree with the venue's withdrawal policy on behalf of myself and my co-authors.